# One-by-One Stainer: A Fast and Hallucination Resilient Domain Adaptation Method for Histopathology

**Karel Moens**[1,2]               KAREL.MOENS@ESAT.KULEUVEN.BE
**Jonas De Vylder**[2]               JONAS.DEVYLDER@BARCO.COM
**Tinne Tuytelaars**[1]             TINNE.TUYTELAARS@ESAT.KULEUVEN.BE
**Matthew B. Blaschko**[1]         MATTHEW.BLASCHKO@ESAT.KULEUVEN.BE
[1] *Dept. of Electrical Engineering, PSI, KU Leuven, Belgium*
[2] *Barco NV, Belgium*

**Editors:** Accepted for publication at MIDL 2026

## Abstract

Histological staining is a crucial step in the analysis of tissue samples, enabling pathologists to identify and diagnose diseases. However, variations in staining protocols and equipment can lead to inconsistencies in image quality. As a result, histological stain normalization remains an active area of research with increasing demand for AI driven diagnosis support. Unlike other domain adaptation problems, in pathology, the consequences of image hallucinations, obscuring or inserting information, are much greater. We propose a method that mitigates the risk of hallucinations by simplifying the network architecture and training process. Our fully $1 \times 1$ convolutional architecture prevents textural modifications and we show that a residual connection combined with weight regularization effectively suppresses color information loss. The model does not need supervision from CycleGAN based models. It is trained directly and leverages target domain color distribution information for better convergence without requiring any paired images. As a result, our method can be trained concurrently on images at varying scales, showing differing anatomical structures or dyes. This simplifies the dataset collection, facilitates the adoption at new centers, and reduces the number of models needed at inference.

Table 1: A comparison of normalization methods with respect to desirable properties for their practical use. For a description of the properties and an elaboration on StainNet, see section 3.

| | StainGAN | StainFuser | ContriMix | StainNet | ours |
|---|---|---|---|---|---|
| Retroactively applicable | ✓ | ✓ | ✗ | ✓ | ✓ |
| Retention of infrequent colors | ✗ | ✗ | ✓ | ✗ | ✓ |
| Resilient to hallucination | ✗ | ✗ | ✗ | — | ✓ |
| Scale independent | ✗ | ✗ | ✗ | ✓ | ✓ |
| Structure independent | ✗ | ✗ | ✗ | — | ✓ |

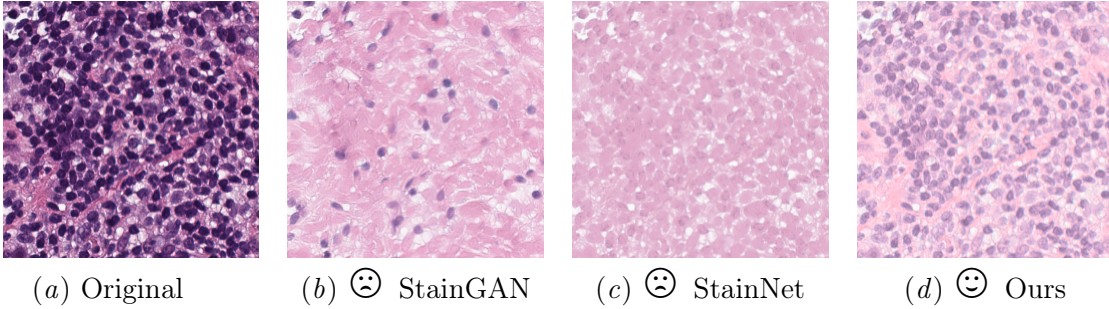

(a) Original      (b) ☹ StainGAN      (c) ☹ StainNet      (d) ☺ Ours

Figure 1: Stain normalization enables consistent visual appearance across histology images from different institutes, enhancing interoperability for downstream models and pathologists. Unlike prior methods that risk hallucinations or structural degradation, our approach preserves tissue integrity and is explicitly designed to be hallucination resilient.

## 1. Introduction

Histopathology plays a critical role in clinical diagnostics, yet it faces persistent challenges in the application of machine learning due to domain shift and data drift. Variability in staining protocols, scanner hardware, and tissue preparation introduces significant heterogeneity across datasets, which can degrade the performance of deep learning models trained on limited or homogeneous data.

As Tellez et al. (2019) emphasize, addressing the variability issues for machine learning requires either data augmentation, normalization, or a combination of both. Vasiljević et al. (2022) caution the use of deep learning methods as they produce varying results and can be prone to hallucinations. This risk is exacerbated with the trend towards larger models with more capacity (Wang et al., 2022; Stegmüller et al., 2023; Alber et al., 2025). These models often function as 'black boxes', prone to regressing toward the most probable modes in the data distribution, potentially overwriting rare but clinically significant features. On the other hand, Swiderska-Chadaj et al. (2020) demonstrate that simple color normalization alone is insufficient to bridge the performance gap across domains. Their findings suggest that more expressive, non-linear models are necessary to capture complex variations such as scanner-specific artifacts and staining concentration differences.

In this work, we propose a lightweight, non-adversarial domain adaptation framework, $1 \times 1$ Stainer, that mitigates hallucination risk and preserves critical histological information. Our method introduces a novel color distribution dissimilarity loss to guide a non-linear color mapping without relying on generative adversarial networks (GANs), cycle consistency or diffusion models. We aim to strike a balance between model expressiveness and interpretability, offering a practical solution for robust histopathological analysis across diverse clinical settings. Our contributions to domain adaptation for histopathology are as follows:

- We propose a method to directly train light-weight domain adaptation models on unlabeled data that are free of hallucinations.

- We propose a color distribution dissimilarity loss function for distribution matching.

- We illustrate the risk of hallucinations in several state-of-the-art deep learning models with examples, and offer an approach to mitigate this risk.

## 2. Related work

Numerous deep learning approaches have been proposed to address domain shift in histopathology (Gangeh et al., 2025). These methods can be broadly categorized into color augmentation and color normalization strategies (Nguyen et al., 2024). Color augmentation increases the diversity of training data by simulating variations in staining (Gao et al., 2022). Color normalization, by contrast, aims to harmonize the appearance of histological images across domains. Recently, normalization has faced criticism for the limited effectiveness of widely used methods. Swiderska-Chadaj et al. (2020) show that simple color normalization fails to fully bridge the performance gap across domains. Moreover, indicators of the data source often remain even after normalization, raising questions about its utility (Dawood et al., 2023; Howard et al., 2021). Such critiques have largely focused on non-deep learning methods. Even if normalization does not completely eliminate domain differences, partial alignment can still yield meaningful gains in downstream tasks. Contrary to augmentation, normalization offers a practical solution for scenarios where retraining models is infeasible, such as when data is unavailable, models are proprietary or third-party, or regulatory approval after retraining is impractical. By harmonizing visual appearance, normalization enables the continued use of existing models. Importantly, augmentations and normalization are not mutually exclusive. Their combined use may offer complementary benefits, augmentations enhance model robustness, while normalization improves interpretability and interoperability, especially in clinical workflows.

The traditional methods for stain normalization such as Reinhard's technique (Reinhard et al., 2001) and its variants, including RandStainNA (Shen et al., 2022), apply linear transformations to match color distributions. However, these approaches may fall short in modeling non-linear variations introduced by different scanners or staining protocols. StainGAN (Shaban et al., 2019) and its derivative StainNet (Kang et al., 2021) employ GAN-based architectures for stain normalization. More advanced methods attempt to disentangle content from style attributes. HistAuGAN (Wagner et al., 2021) and ContriMix (Nguyen et al., 2024), for example, use adversarial objectives to extract attribute and content encodings. These can then be swapped and mixed for augmentations. While seemingly visually effective, adversarial training suffers from high sample complexity and training instability (Gulrajani et al., 2017; Jabbar et al., 2021; Wiatrak et al., 2019), and can be prone to hallucinating features, as shown by Vasiljević et al. (2022).

Recently, diffusion models such as StainDiff (Shen and Ke, 2023) and StainFuser (Jewsbury et al., 2024) have emerged as alternatives to GANs. These models add noise to input images and regenerate them in a target style, using cycle consistency losses to preserve structural integrity. However, the denoising process inherently risks overwriting subtle but important features, raising concerns about their reliability in clinical applications.

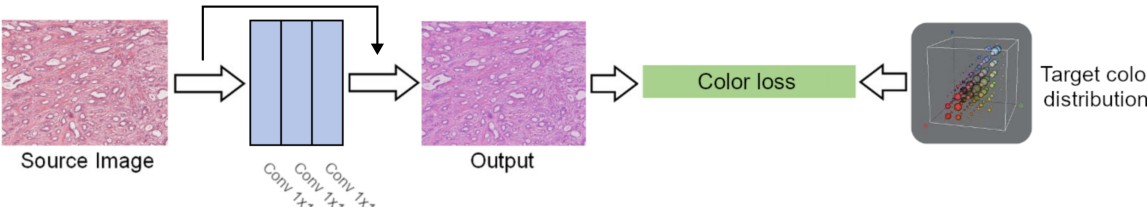

Figure 2: To prevent hallucinations, the network is constrained to only three 1×1 convolutional layers. This limits the non-linear behavior of the functions that can be expressed. The added skip connection and regularization on the weights keep the learned function smooth and close to the identity. Within these constraints, the network is trained in an unsupervised fashion to match the color distribution of the target domain.

## 3. Practical properties

The similarity of the output images to the target domain and the performance of downstream applications after normalization are important measures for comparing stain normalization methods and are discussed in Section 5. Other notable properties include the ease of training and the speed of inference. Due to the simplicity of the loss function and network architecture, our method excels in these aspects, see the appendix for details. Besides those, there are other desirable properties for normalization methods, as listed in Table 1.

### Retroactively applicable

Although data augmentation may improve inference speed compared to normalization, it is often impractical to generate a newly augmented version of the training dataset when novel methods or data become available. This process would require access to the original training data and the training algorithm used for the downstream task. Moreover, it could invalidate existing regulatory approvals for the downstream method, necessitating a repeat of the approval process, which may include conducting new trials. However, stain normalization does not alter the downstream method. As a result, it is easier to train a normalization method in an unsupervised manner and have this simpler step cleared to improve the performance of downstream analysis. Stain normalization is retroactively applicable to methods that have already been developed and approved, possibly by a third party.

### Retention of infrequent colors

A form of information loss occurs when multiple colors are mapped to the same output. We see this in StainNet, for example when it is tasked with mapping colors that are not well represented in its training dataset, Figure 10 in the appendix. This happens to the yellow-brownish specks in Figure 3. Our networks contain residual skip connections and are trained with weight regularization. This keeps the learned mapping close to the identity function.

RESILIENT TO HALLUCINATION

Whenever a typical neural network is tasked with generating content, there is a risk of hallucination. In the case of stain normalization, we define hallucinations as the introduction or removal of structures in the image. Our method avoids such hallucinations as the entire processing windows are limited to one pixel at a time. This makes the generation of new structures impossible. An analogous argument can be made for StainNet, which has a very similar architecture with only 1×1 convolutions but without the skip connection. As illustrated in Figure 1, in practice, we do observe the removal of clinically relevant structures in StainNet outputs. Our hypothesis is that this is due to the direct supervision by StainGAN, which does not have the same limitation and can hallucinate structures. When a supervising StainGAN model outputs several colors for the same input color, depending on the context, the distilled StainNet model is forced to average these colors. It seems that StainNet models are particularly susceptible to this since they are prone to discarding color information, as discussed in the previous paragraph.

SCALE INDEPENDENT

Most stain normalization methods take crops at a specified zoom level as input. Even simply cropping and stitching the results back together can produce inputs at the wrong scale and therefore unexpected results. Our method functions only on colors and is unaffected by the scale at which those colors are presented. Even for training, heterogeneous scales may be present within and between the source and target dataset. This makes it easier for practitioners to construct a dataset.

STRUCTURE INDEPENDENT

When the receptive field of a method is limited to one pixel at a time, it is impossible to rely on anatomical structures for the generation process. All such methods do is model the difference between the digitization of true colors due to factors such as slide preparation, scanner, and settings. As such, it does not matter what anatomical structure or applied protein exhibits that color. This allows a multimodal approach to constructing datasets for training, as slides from different sources can be aggregated.

## 4. Method

Stain normalization seeks to transform histopathological images such that those acquired through one preprocessing pipeline and scanner setup resemble images processed through a different pipeline. Crucially, this transformation models only the digitization process, the underlying tissue slide remains unchanged. Given that the most prominent inter-domain differences are typically in color, while texture and structural features originate from the tissue itself, we conceptualize stain normalization as a color flow, a deterministic function operating on pixel values. By design, this function is spatially invariant, meaning it does not adapt its output based on local image context. This constraint is intentional, as allowing spatially dependent transformations could introduce hallucinations, where the model fabricates plausible but incorrect features.

### 4.1. Convolutions of 1×1 with limited non-linearities

As in other medical fields, in histopathology, it is paramount that no information is lost or added as fabrications. What seems like minor details in a big picture could have a big impact on the diagnosis. To achieve this information retention, we chose to constrain the space of functions that can be expressed with the network architecture. Taken to its extreme, a reversible linear transformation between colors is by nature immune to hallucinations. Of course, in general, such a color shift can by itself not adequately counteract the differences between images scanned at different wet labs. For example, a difference in staining concentration would result in a different hue for the tissue but should not affect the background.

We therefore took linear color transformations as a starting point. Following common practices and architectures, we then incrementally explored the design space to arrive at a network that achieves good results in various settings. This translates to fully convolutional neural networks, exclusively composed of kernels with spatial sides of $1 \times 1$. The number of layers and the number of kernels per layer is kept low, up to three convolutional layers with up to 32 kernels per convolution. Additionally, a residual approach is taken. The output of the neural network is added to the input colors instead of serving as the final result. This allows regularizers on the size of the network weights to keep the learned function closer to the identity. Beneficial byproducts of these design decisions are a low memory footprint and fast execution.

### 4.2. Color distribution dissimilarity loss

The most noticeable gap between domains in histopathology is the overall hue and occurrences of colors. Closing this gap in color distributions is a necessary condition for adapting one domain to the other. We therefore propose to incorporate the color likelihoods in the loss function to guide the optimization of the model. In contrast to Lee and Ma (2022), we do not introduce another neural network but derive losses directly from the occurrences in the target dataset and batches.

Earth mover's distance is a common metric for distribution similarity. As colors are represented by vectors of size three, aggregating their occurrences in a dataset gives rise to a three-dimensional tensor. Calculating the earth mover's distance between such tensors would be iterative, approximate and costly. Instead, we propose to optimize an upper bound on this distance. In a 1D space, the earth mover's distance can be calculated efficiently as the cumulative sum of the differences. The sum over the distances of each color channel independently gives an upper bound over the distance between the full distributions, as the transport can be redundant and not as optimal as when all dimensions are considered concurrently. This approximation by projecting along the RGB unit vectors is somewhat arbitrary. We therefore take the mean over the upper bounds set by $N$ random orthonormal bases, sampled uniformly on the unit sphere, similar to Seguy et al. (2018). The projected distribution dissimilarity loss function $L_{\text{color,proj}}$ can be defined as follows:

$$L_{\text{color,proj}} = \frac{1}{N} \sum_{i=1}^{N} \sum_{j=1}^{3} \text{OT}(H_{\text{proj},i,j}, H_{\text{target,proj},i,j}) \tag{1}$$

where $N$ is the number of random orthonormal bases, $H_{\text{proj},i,j}$ is the 1D histogram of the batch projected along the $j$-th vector of the $i$-th basis, in practice differentiably approximated

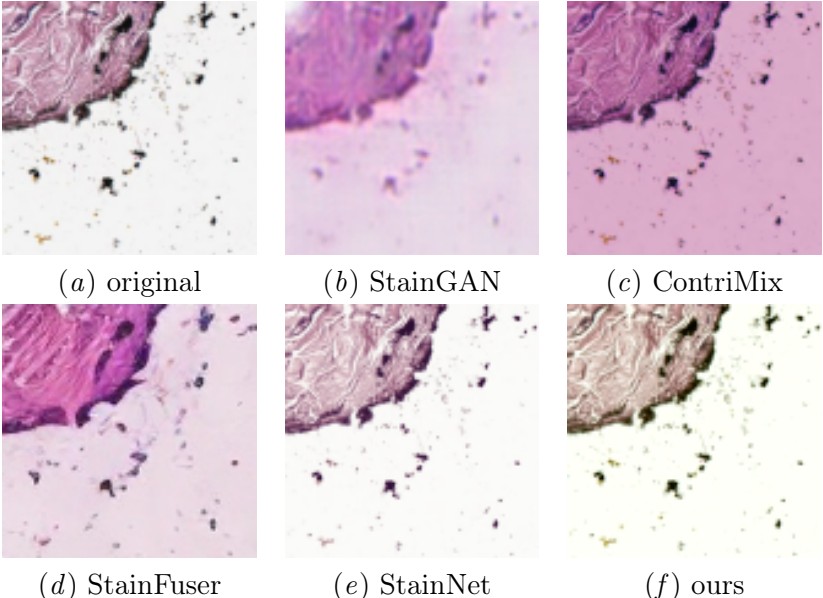

(*a*) original      (*b*) StainGAN      (*c*) ContriMix

(*d*) StainFuser      (*e*) StainNet      (*f*) ours

Figure 3: Normalization by different models trained for CAMELYON16 data. To illustrate the risks when used out of distribution, the input image is taken from our proprietary dataset. StainGAN and StainFuser show hallucinations in the upper left corner. Contrast is reduced by ContriMix. StainNet maps the yellow specks to similar purples as in the rest of the image. Full versions in the Appendix.

as proposed by Ustinova and Lempitsky (2016), $H_{\text{target,proj},i,j}$ is the 1D histogram of the target distribution projected along the $j$-th vector of the $i$-th basis, and OT denotes the optimal transport cost or earth mover's distance between the projected histograms. The model architecture and training procedure is illustrated in Figure 2.

## 5. Results

The proposed method is evaluated on four datasets: the Mitos & Atypia 14 dataset (Roux, 2014), the CAMELYON16 dataset (Bejnordi et al., 2017), the MIDOG 2021 dataset (Aubreville et al., 2023), and a proprietary, real-world dataset, see Appendix B. The first contains coregistered images from two different domains. This allows the computation of several image-to-image metrics. Downstream classification is evaluated on the second dataset. The latter dataset was constructed to investigate occurrences of hallucinations. Figure 1 and Figure 3 show images from our proprietary dataset. They illustrate some of the risks involved with the use of state-of-the-art stain normalization methods. For Figure 1, new models were trained on the proprietary data, using the code and hyperparameters provided by the original authors. Despite this, the StainGAN and StainNet models display clinically significant hallucinations. There is a clear reduction of identifiable nuclei and, overall, a shift toward an appearance in line with treatment effect. The count of so-called ghost cells

is not preserved from input to output images, which could profoundly mislead pathologists and alter the grading and interpretation from benign to necrotic.

Table 2: Paired image metrics for the test set of the Mitos & Atypia 14 dataset. The $\Delta$ columns list the improvements over applying the identity transformation. The column on the right is the Kullback-Leibler divergence over the colors. It does not list deviations as the divergence between the output and the target distribution is a single value. The deviations in the SSIM and PSNR columns indicate that there is no clear best method for these metrics, as the error margins overlap for all approaches. The $SSIM_{src}$ is a measure of the preservation of information from the source image, a crucial property for medical diagnosis, see section 5.1.

| Method | SSIM ($\uparrow$) | $\Delta$ SSIM | PSNR ($\uparrow$) | $\Delta$ PSNR | $SSIM_{src}$ | KL ($\downarrow$) |
|---|---|---|---|---|---|---|
| source | $.641 \pm .108$ | $0.0$ | $20.3 \pm 3.2$ | $0.0$ | $1.0$ | $1.85$ |
| Reinhard | $.617 \pm .106$ | $.001 \pm .028$ | $19.9 \pm 2.1$ | $2.16 \pm 2.65$ | $.964 \pm .031$ | $0.528$ |
| Macenko | $.656 \pm .115$ | $.026 \pm .019$ | $20.7 \pm 2.7$ | $1.57 \pm .91$ | $\mathbf{.966 \pm .049}$ | $1.31$ |
| Vahadane | $.664 \pm .116$ | $.029 \pm .022$ | $21.1 \pm 2.8$ | $1.81 \pm .93$ | $\mathbf{.967 \pm .046}$ | $1.70$ |
| StainDiff[a] | $.721 \pm .017$ | | | | | |
| StainGAN | $.706 \pm .099$ | $\mathbf{.061 \pm .028}$ | $22.4 \pm 2.6$ | $2.11 \pm 1.24$ | $.883 \pm .025$ | $0.095$ |
| StainNet | $.691 \pm .107$ | $\mathbf{.050 \pm .013}$ | $22.5 \pm 3.3$ | $2.22 \pm .65$ | $.957 \pm .007$ | $1.04$ |
| 1×1 Stainer | $.651 \pm .107$ | $.010 \pm .005$ | $22.0 \pm 3.4$ | $1.73 \pm .58$ | $\mathbf{.997 \pm .0003}$ | $0.200$ |

[a]Results are taken from the original paper.

Table 3: The performance of a set of pretrained tumor-normal classifiers on the test set of CAMELYON16 after input normalization. For testing, the input images originate from the source domain, transformed by the listed methods. $AUC_{retro}$ shows the performance when the normalization is retroactively applied to the classifier that performed best in the source domain. $AUC_{best}$ lists the best observed classifiers.

| Method | Precision | Recall | AUC | $AUC_{retro}$ | $AUC_{best}$ |
|---|---|---|---|---|---|
| source | $.651 \pm .032$ | $.974 \pm .013$ | $.723 \pm .034$ | $.794$ | $.794$ |
| source augmented | $.829 \pm .023$ | $.820 \pm .033$ | $.824 \pm .011$ | NA | $.849$ |
| Vahadane | $.892 \pm .026$ | $.746 \pm .045$ | $.827 \pm .011$ | $.823$ | $.847$ |
| StainGAN | $.902 \pm .027$ | $\mathbf{.879 \pm .020}$ | $\mathbf{.890 \pm .011}$ | $.890$ | $.897$ |
| StainNet | $\mathbf{.952 \pm .020}$ | $.813 \pm .026$ | $\mathbf{.885 \pm .005}$ | $.880$ | $.893$ |
| ContriMix | $.807 \pm .030$ | $\mathbf{.865 \pm .031}$ | $.828 \pm .011$ | $.835$ | $.842$ |
| 1×1 Stainer (ours) | $\mathbf{.937 \pm .016}$ | $.812 \pm .023$ | $\mathbf{.878 \pm .004}$ | $.880$ | $.898$ |

### 5.1. Stain transfer results

Table 2 lists image similarity results for several histopathology domain adaptation methods. The numbers for Reinhard, Macenko, and Vahadane are as reported by the authors of StainNet. To evaluate the performance, the similarity is measured between output images and the matching images from the target domain. To counter the high variance over the images in the test set, the improvement over doing nothing, i.e. the similarity between source

and target images, is also reported under $\Delta$ SSIM and $\Delta$ PSNR. As was done for StainNet, we list $\text{SSIM}_{src}$ as a measure of the preservation of source image texture. It is calculated as the SSIM between the input image in grayscale and the output image in grayscale. The final column on the right shows how well the target color distribution is matched. For this, the discrete color distributions are put in a total of $32 \times 32 \times 32$ bins each containing $8 \times 8 \times 8$ RGB colors and the Kullback-Leibler divergence is calculated between the distribution over output images and the target distribution.

Our method is competitive with others with respect to similarity. A number of samples are included in the appendix for qualitative comparison, see Figure 13. Note that paired pathology data always involves two physical acts of scanning, resulting in differing artifacts, such as focal planes and blur. It may not be desirable to perfectly reproduce these variations. On the other hand, if the input image suffers from artifacts, our method will not cover up its poor quality (Figure 12 in the appendix). Such transformations could lead to hallucinations. 1×1 Stainer matches the target color distribution well, though not as closely as StainGAN, which is not restricted to color mappings and therefore not as influenced by the color occurrences in the source domain. To further illustrate the color matching, Figure 9 in the appendix compares RGB histograms. Finally, $\text{SSIM}_{source}$ shows that 1×1 Stainer outperforms other methods in terms of structural preservation, or absence of hallucinations.

## 5.2. Downstream tumor classification

Following evaluations described in StainNet, we train a normalization model on CAME-LYON16. We then evaluate the effect on a set of 20 classifiers pretrained on the source train set. Vahadane (Vahadane et al., 2016) is taken as a representative method for non-deep learning based approaches, as previous works have shown its superiority over others in downstream classification (Shaban et al., 2019). The line 'source' is the baseline without normalization. As another baseline, 'source augmented' contrasts normalization with applying color jitter during the training of the classifiers. $\text{AUC}_{retro}$ reflects a realistic deployment scenario: it measures the performance improvement when normalization is retroactively applied to the best-performing classifier trained on unnormalized source data. $\text{AUC}_{best}$, on the other hand, represents the upper bound of performance observed in our experiments. It lists the best-performing classifier for each normalization method. Our model is among the best in our test. It is competitive with StainGAN and StainNet, with overlapping error margins.

Upon visual inspection of samples that are misclassified through our method but correctly processed after normalization by StainGAN or StainNet, a pattern was observed. Many of the mistaken source images contain unnatural colors as scanning artifacts (Figure 12 in the appendix). As by design, our method retains the rarity of these colors and this might have lead to the confusion of the classifier. By contrast, StainGAN and StainNet paint over these anomalous pixels with common colors. In this way, they suffer from color information loss. Even though, in these cases, artifacts are modified, in our opinion, stain normalization and domain adaptation models are not trained to distinguish between artifacts and rare clinically relevant colors. The responsibility of quality assurance is better left to dedicated models and expert human supervision. The potential marginal downstream gains from allowing normalization models to modify anomalous pixels and patches to what's expected does not outweigh the risk of masking out rare semantically relevant information.

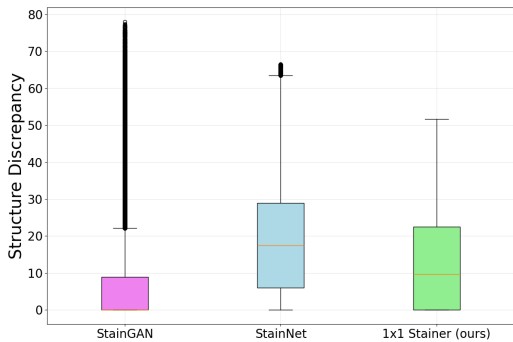 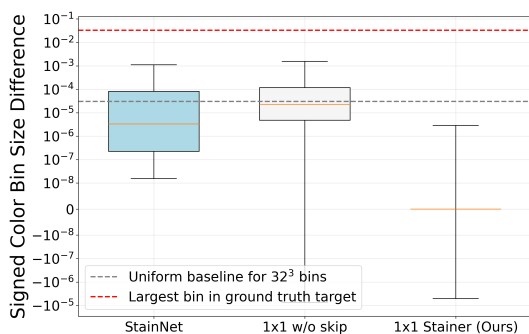

(a) Hallucinations are hard to define. Structure Discrepancy is a measure to identify potential hallucinations. See Table 5 in the Appendix for examples. Importantly, this experiment indicates a risk of higher discrepancies with StainGAN and StainNet.

(b) These box plots show how color rarity shifts after normalization. A positive value means that an infrequent color is mapped to a more common one. StainGAN can not be compared against as it requires the surrounding pixels. 1×1 w/o skip is an ablation.

Figure 4: Quantitative results for hallucination resilience and infrequent color retention.

## 5.3. Quantitative measures of hallucination

Structure Discrepancy (Moens et al., 2026) is a recently proposed method to quantify hallucinations in histopathology images. Based on edge detection, high values suggest that a structure was removed or inserted. Though impossible to perfectly align with medically salient hallucinations, especially at lower values, see Table 5, it is effective at finding examples of unwanted modifications. Due to the computational and time costs of retraining methods on large, challenging datasets, we are forced to limit the comparison to StainGAN and StainNet. A more extensive quantitative review of hallucinations in state-of-the-art stain normalization is left for future work. Figure 4(a) shows that our model does not produce outlier discrepancies, with lower observed maximum values in our 24k test set. This empirically validates that our method avoids introducing rare yet clinically significant hallucinations.

To quantify the retention of infrequent colors discussed in Section 3, we compared the relative occurrences of colors in the source domain to their occurrences in the target domain after normalization. This was done uniformly over all RGB colors, to give as much weight to an outlier color as to an expected one. In Figure 4(b) our method is nicely centered around and concentrated at zero. On the other hand, StainNet tends to map infrequent colors to more common ones, resulting in a loss of information. Given their similar construction, see Appendix A, this shows that our method brings all the advantages of StainNet while improving on hallucination resilience.

Table 4: Ablation study results on CAMELYON16 downstream classification. Rows compare the impact of weight regularization (w/o reg), residual skip connection from input to output (w/o skip), and network depth (1 layer, 6 layer) against the proposed 3-layer architecture.

| Method | Precision | Recall | AUC | $\mathbf{AUC}_{retro}$ | $\mathbf{AUC}_{best}$ |
|---|---|---|---|---|---|
| 1×1 Stainer (ours) | **.937 ± .016** | .812 ± .023 | **.878 ± .004** | .880 | .898 |
| 1×1 Stainer w/o reg | .891 ± .015 | .774 ± .021 | .839 ± .006 | .842 | .852 |
| 1×1 Stainer w/o skip | .854 ± .017 | .797 ± .018 | .830 ± .009 | .833 | .844 |
| 1×1 Stainer 1 layer | **.944 ± .015** | .789 ± .030 | .870 ± .008 | .873 | .892 |
| 1×1 Stainer 6 layer | **.969 ± .010** | .738 ± .036 | .857 ± .013 | .853 | .906 |

## 6. Ablation studies

To better understand the contributions of different components in our method, we conduct ablation studies examining the impact of weight regularization, network depth, and training data distribution on downstream classification performance.

### 6.1. Weight regularization

As an ablation, we trained our model without regularization on the network weights. Only the color distribution dissimilarity was used for the loss function. Without this regularization, we expect the learned color mapping to be less smooth and differ more from the identity mapping. The impact on downstream classification is shown in Table 4. The performance of the model without regularization dropped compared to the regularized version, as seen in the lower precision, recall, and AUC values. However, it remains on par with the source augmented baseline. When the model is furthermore trained without the skip connection, the convergence becomes less stable. The training process risks ending on local minima with mappings akin to color inversions.

### 6.2. Network depth

To investigate the impact of network depth on performance, we trained variants with 1 layer and 6 layers, compared to the default 3-layer architecture. The results are summarized in Table 4.

### 6.3. Training data distribution

The convergence of our color distribution matching is dependent on the content of the source and target datasets used during training. It works best when the same physical true colors are digitized just as often in both domains. For example, this condition is trivially satisfied in a dataset of paired images. The weight decay regularization incentivizes smooth mappings close to the identity. This ensures that the occurrences do not need to match exactly.

A setting in which a mismatch in colors could occur is when the source and target domains contain different tissue types. To test the applicability in such a scenario, we trained a model on a heterogeneous set of tissue slides from our proprietary data to normalize to the Mitos & Atypia 14 target domain for qualitative evaluation. The set contained 34 brain slides,

1030 lung slides, 514 pancreas slides, 287 skin slides, 842 uterus slides, and importantly does not include any breast tissue slides. A few random examples of the diversity are show in Figure 14 in the appendix. Qualitative comparisons to test target patches are included in Figure 15. Even in this challenging setting, our method retains its high SSIM$_{src}$ of .997 and has a PSNR of $21.3 \pm 3.1$, still with largely overlapping error margins. It should be noted that GAN or diffusion based models would, without strong conditioning, most likely fail in this setting. Even with cycle consistency, the differences between image content in the domains would incentivize such model to alter the content of inputs.

## 7. Limitations

Our method learns a color transformation function that models the preparation and digitization process of a specific scanning pipeline. This approach is both structure and scale independent, as the mapping operates solely on pixel colors without regard to anatomical features or image resolution. The method does not address domain differences beyond color, such as variations in texture or structural features introduced by different scanners or sample preparations. It is designed to close the domain gap through color normalization, by design to prevent hallucinations.

## 8. Conclusion

We introduced 1×1 Stainer, a novel method for normalization between histopathology domains. Our results show that our design decisions are, above all else, effective in mitigating hallucinations and information loss during processing of medical images. The training procedure is unsupervised and by avoiding a cycle consistency loss, more straightforward than similar state-of-the-art methods. The proposed color distribution dissimilarity loss leads to models that closely match the color characteristics of the target distribution. Our approach has several desirable properties that facilitate its application.

## Acknowledgments

This research is funded by VLAIO through the LEASTwork project (HBC.2021.0228).

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

## Appendix A. Implementation and computational performance

The network architecture is the same as in StainNet, i.e. a fully $1 \times 1$ convolutional neural network with three layers of 32 kernels each, followed by a new residual addition at the end. Adam optimizers are used with a learning rate of 0.0001 and batches of 8 images. For every batch, $N := 8$ orthonormal bases are sampled uniformly for the color distribution dissimilarity loss. The target color distribution for the dissimilarity is aggregated by sweeping over the target train set. An L1 weight decay loss term is added with $\lambda_{reg} := 0.25$. In our implementation, this comes down to 386 MiB of GPU memory usage while idle. During inference on a single NVIDIA RTX 4090 GPU the memory requirements rise to 2500

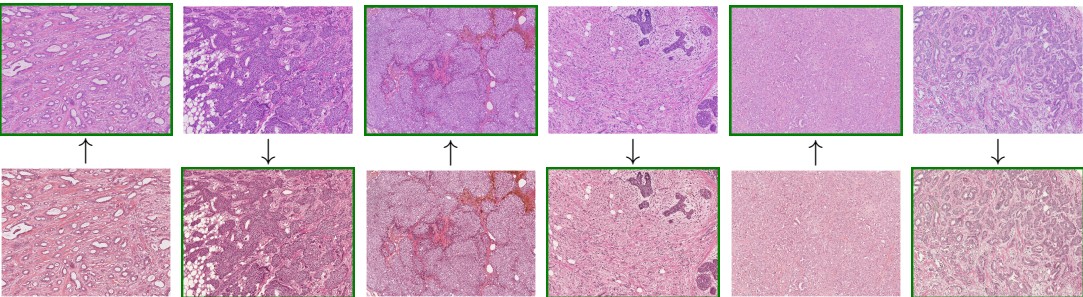

Figure 5: Six samples from the MIDOG 2021 dataset and their transformations by the respective 1×1 Stainer models are shown. Images with a green border are the modified version of the image in the same column. The rows represent two different domains in the dataset. Note that MIDOG 2021 does not contain image pairs.

MiB for streaming large images, i.e. batches of $1 \times 3 \times 4096 \times 4096$. These images are processed with a delay of $.0350s \pm .0019s$ or $28.62 \pm 0.41$ fps.

## Appendix B. Hallucination evaluation dataset

To study the risk of hallucinations, we collected a proprietary dataset of real-world data from multiple institutes. The sources of data include The Cancer Imaging Archive (TCIA, Clark et al. (2013)), Michigan Medicine, and Charité – Universitätsmedizin Berlin. From these, we extracted subsets containing only breast tissue samples, with only one scanner model per domain. While future work will examine heterogeneous datasets with multiple tissue types, here we limit the study to breast tissue only as many related works are designed for a homogeneous setting. The source set originated from TCIA and contains 65k patches scanned through a Leica Aperio AT2. The target set of 18k patches comes from Michigan Medicine and was scanned using a Leica Aperio GT 450 DX. Finally, a test set of 24k patches scanned with the same model of scanner as in the source set was collected from Charité. Beyond tissue type and scanner model, these subsets were not curated. Upon visual inspection and annotation, several artifacts were discovered, as would be the case in an actual clinical practice. These artifacts include air bubbles, tissue folds, misaligned image stitching, markers, overstaining, foreign materials, and out-of-focus regions. For comparison, the CAMELYON16 dataset (Bejnordi et al., 2017) is far more sterile and is limited to 6k images in its source and target domains each, and 10k in the test set.

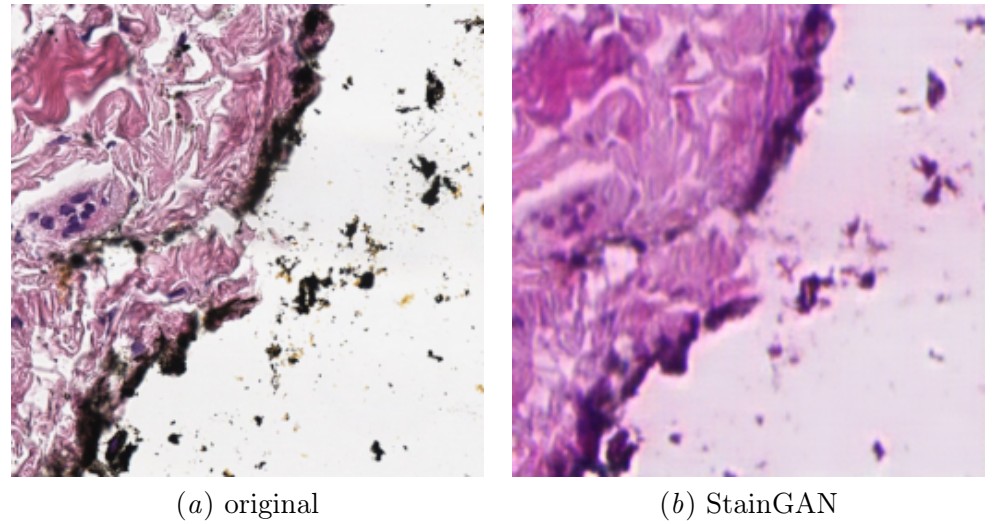

(*a*) original                    (*b*) StainGAN

Figure 6: Stain normalization examples.

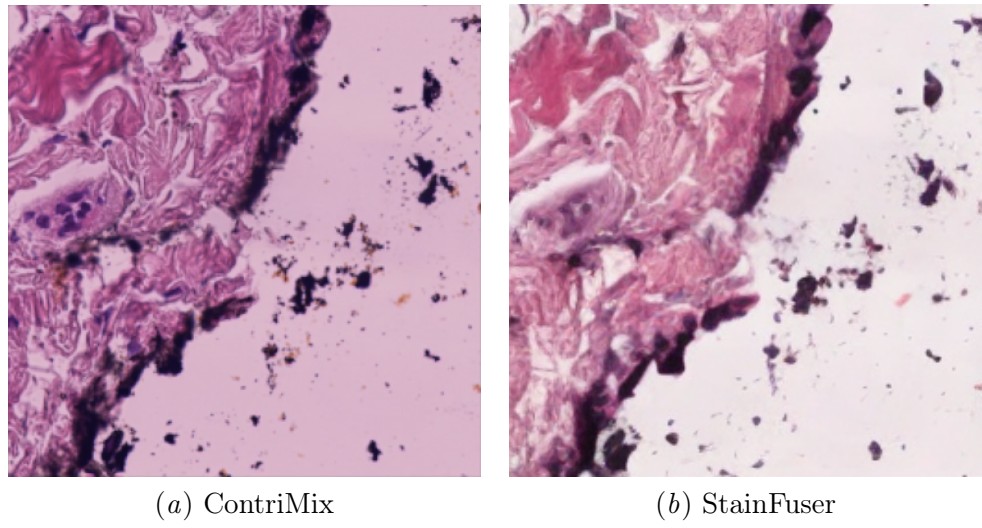

(*a*) ContriMix                    (*b*) StainFuser

Figure 7: Stain normalization examples.

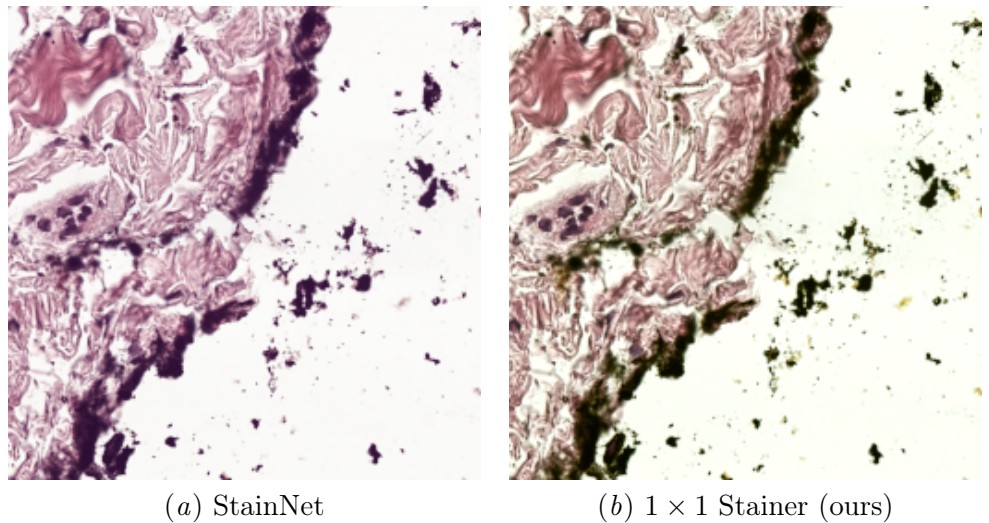

(*a*) StainNet          (*b*) $1 \times 1$ Stainer (ours)

Figure 8: Stain normalization examples.

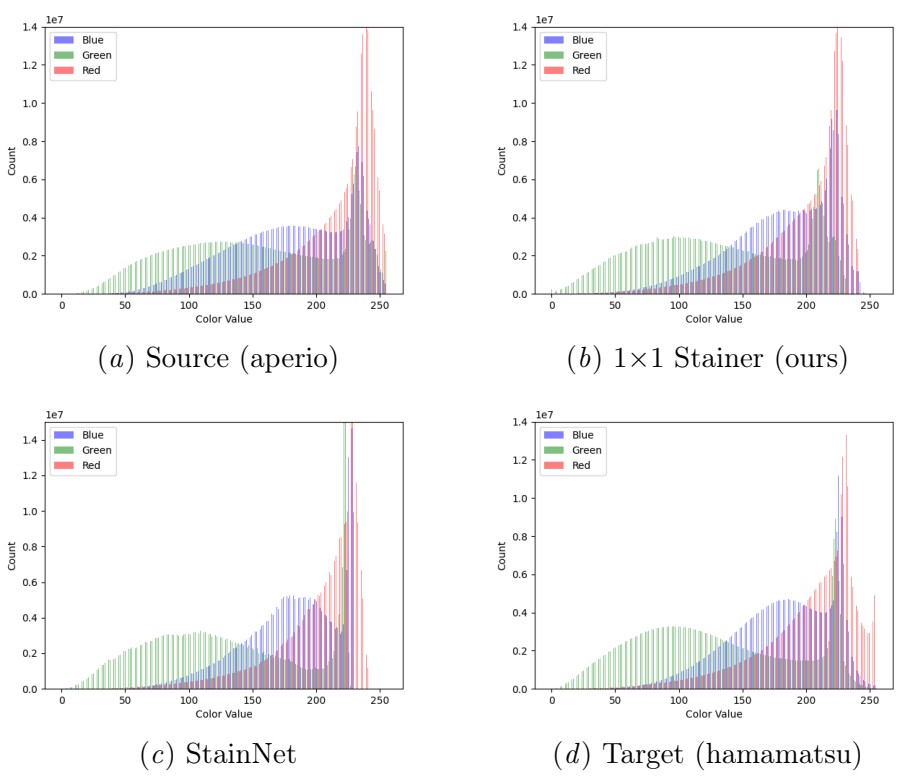

Figure 9: The RGB color distributions after the source is modified to resemble the target by 1×1 Stainer and StainNet (Kang et al., 2021) are shown.

Table 5: Examples of Structure Discrepancy ([Moens et al., 2026](#)). Even when scores are close, the perceived hallucinations can vary in severity, especially at the lower end.

| | Input | Output | Structure Discrepancy |
|---|---|---|---|
| StainGAN | | | 48 |
| | | | 70 |
| StainNet | | | 46 |
| | | | 64 |
| 1×1 Stainer | | | 49 |
| | | | 50 |

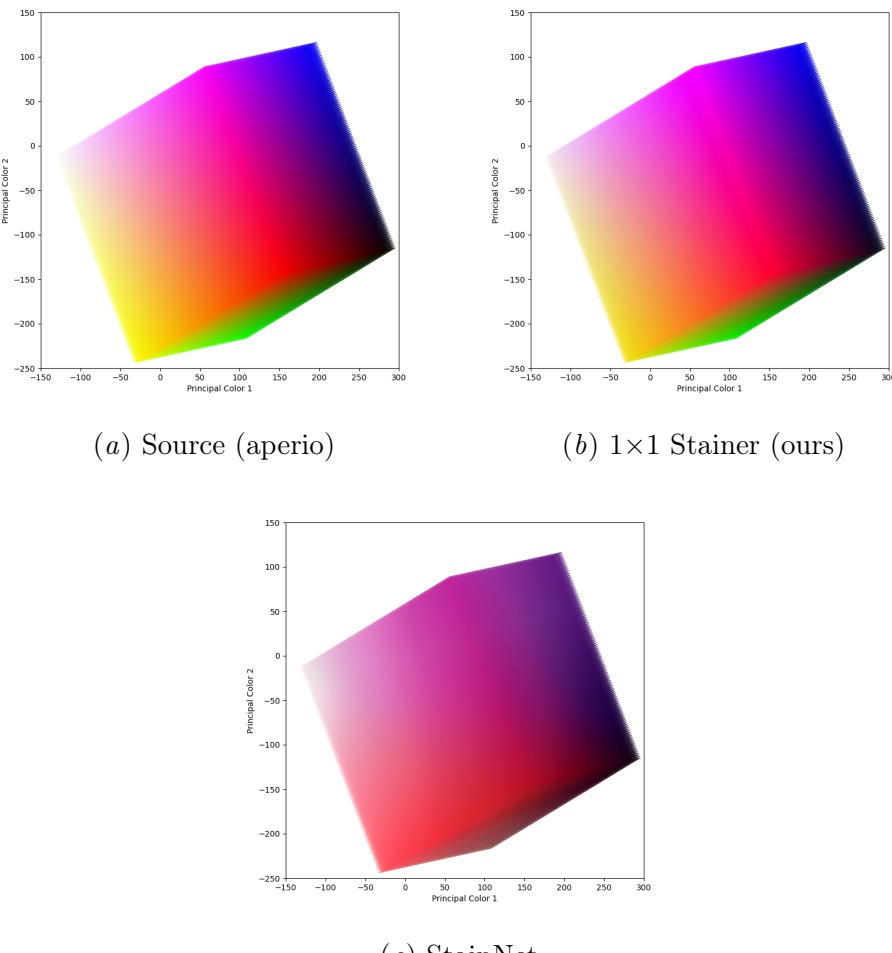

(*a*) Source (aperio)          (*b*) 1×1 Stainer (ours)

(*c*) StainNet

Figure 10: An illustration of the color mapping by 1×1 Stainer and StainNet. (a) A projection is taken of the RGB color cube perpendicular to the principal component vectors of the set of colors in the source train set of Mitos & Atypia 14. (b) At the same position on the projection, the colors transformed by our method are shown. The colors appear shifted and warped. Notice how there is barely any orange past 50 times the first principal component vector. (c) StainNet maps everything to colors that are expected in the target distribution. In this way, information from out of distribution colors might be lost.

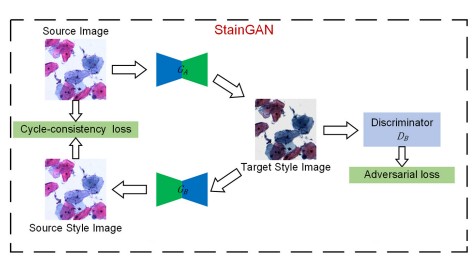

(*a*) The StainGAN (Shaban et al., 2019) training process involves an adversarial cycle-consistency loop with two generator networks and a discriminator.

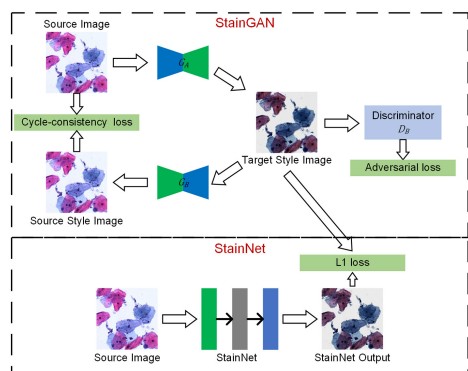

(*b*) The StainNet (Kang et al., 2021) training process first requires training of a StainGAN model for supervision.

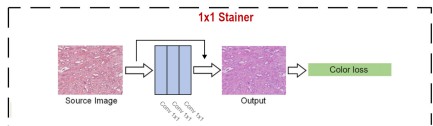

(*c*) During the 1×1 Stainer training process (ours) a residual network is supervised directly by the target color distribution.

Figure 11: A bird's-eye view comparison of the training process for StainGAN, StainNet, and 1×1 Stainer. StainNet and 1×1 Stainer have an extremely small number of parameters, which bounds the modifications that can be made to the input images. These diagrams highlight the difference in simplicity of training, as well as the dependency of StainNet on StainGAN. Figure (b) is reproduced exactly as in the original publication (Kang et al., 2021) and (a) is adapted from it.

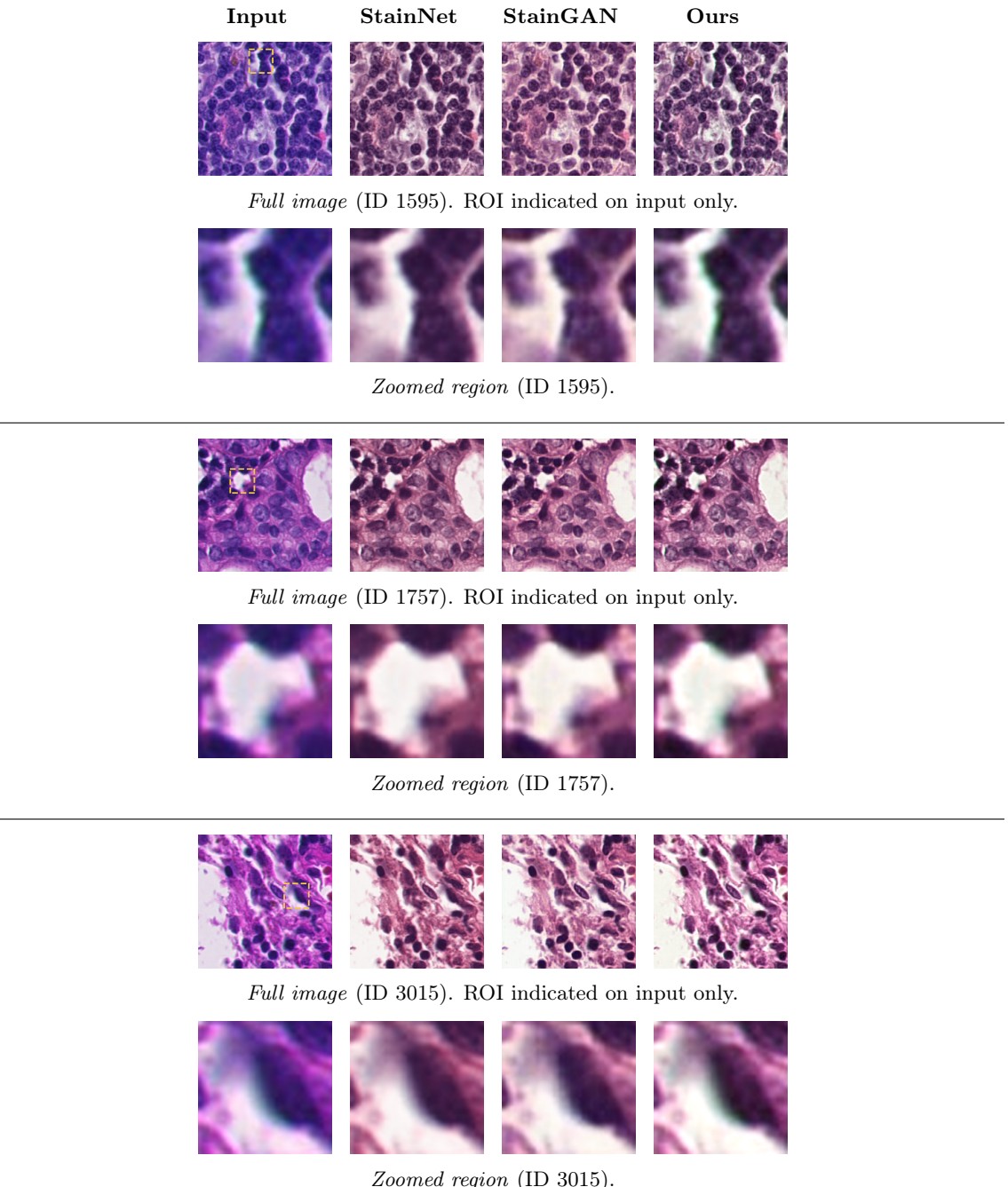

Figure 12: Qualitative comparison of CAMELYON16 misclassifications. Columns show the *Input*, two related works (*StainNet*, *StainGAN*), and *1×1 Stainer (Ours)*. For each example, the top row displays the full image (ROI indicated with a gold dashed box on the input only), and the bottom row shows the corresponding zoomed region. Our method retains infrequent colors to avoid information loss.

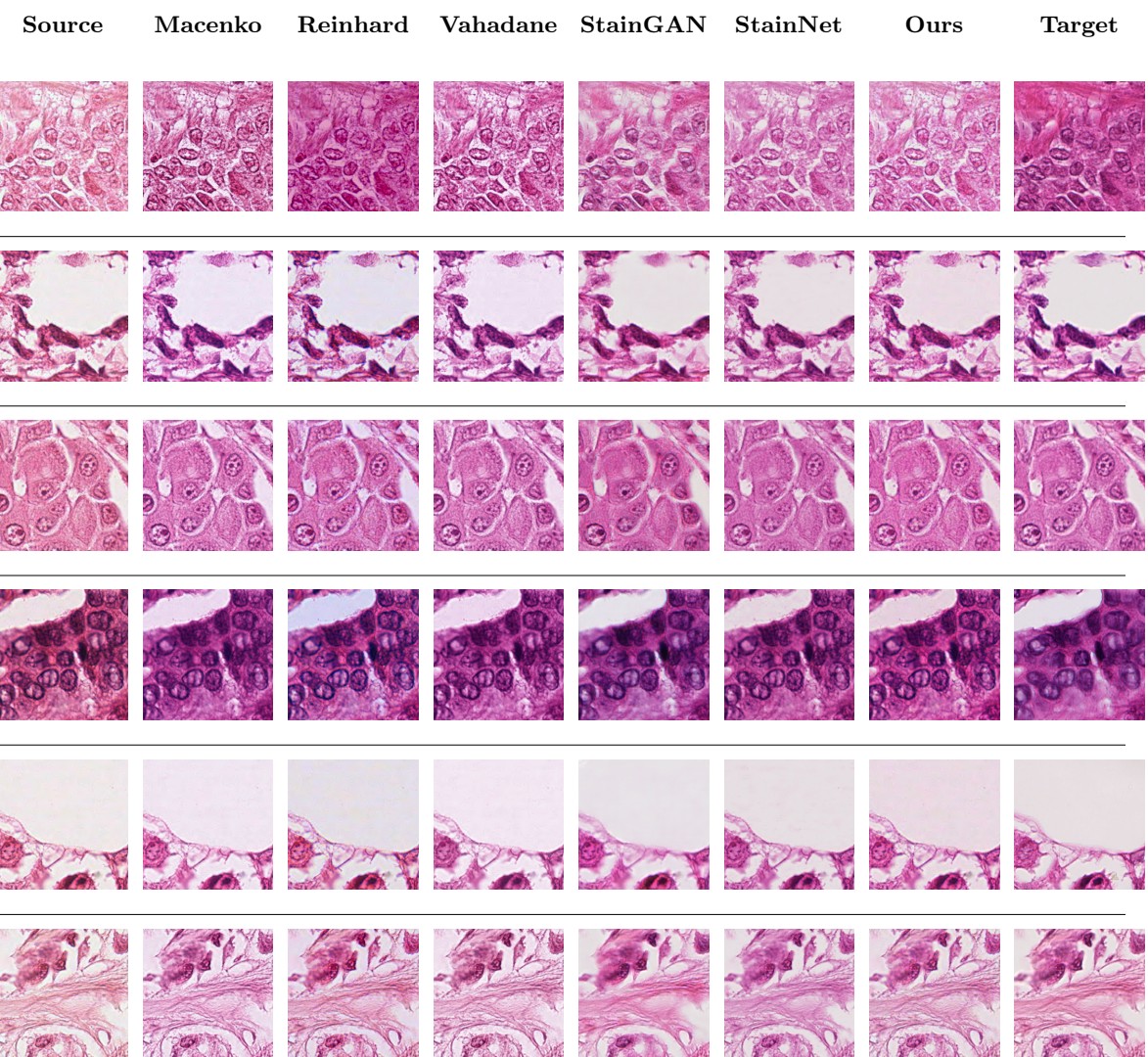

Figure 13: Qualitative comparison across many methods on Mitos & Atypia 14. Columns show the *Source* input (Aperio), classical stain normalizations (Macenko, Reinhard, Vahadane), learning-based methods (StainGAN, StainNet), *1×1 Stainer (Ours)*, and the ground truth *Target* (Hamamatsu).

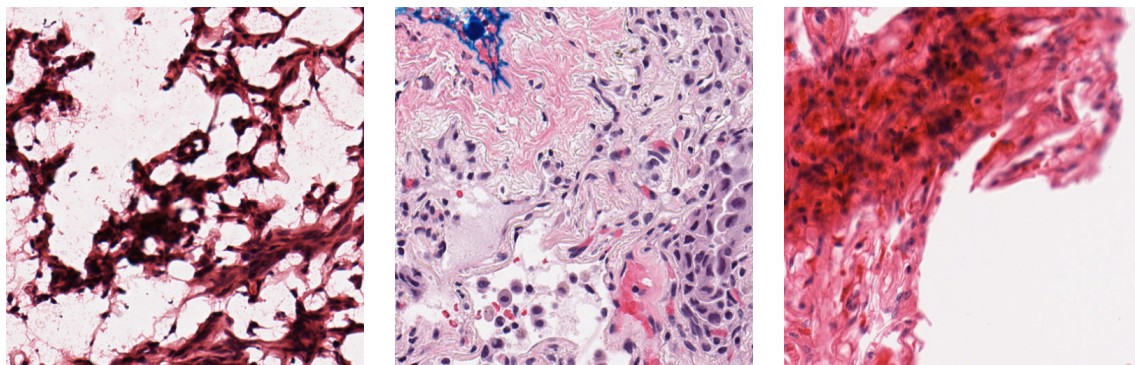

Figure 14: Random samples from the training source dataset used during a study on the applicability of our method on datasets with heterogeneous tissue types. See section 6.3 for details.

| **Source** | **Ours** | **Target** |
|:---:|:---:|:---:|

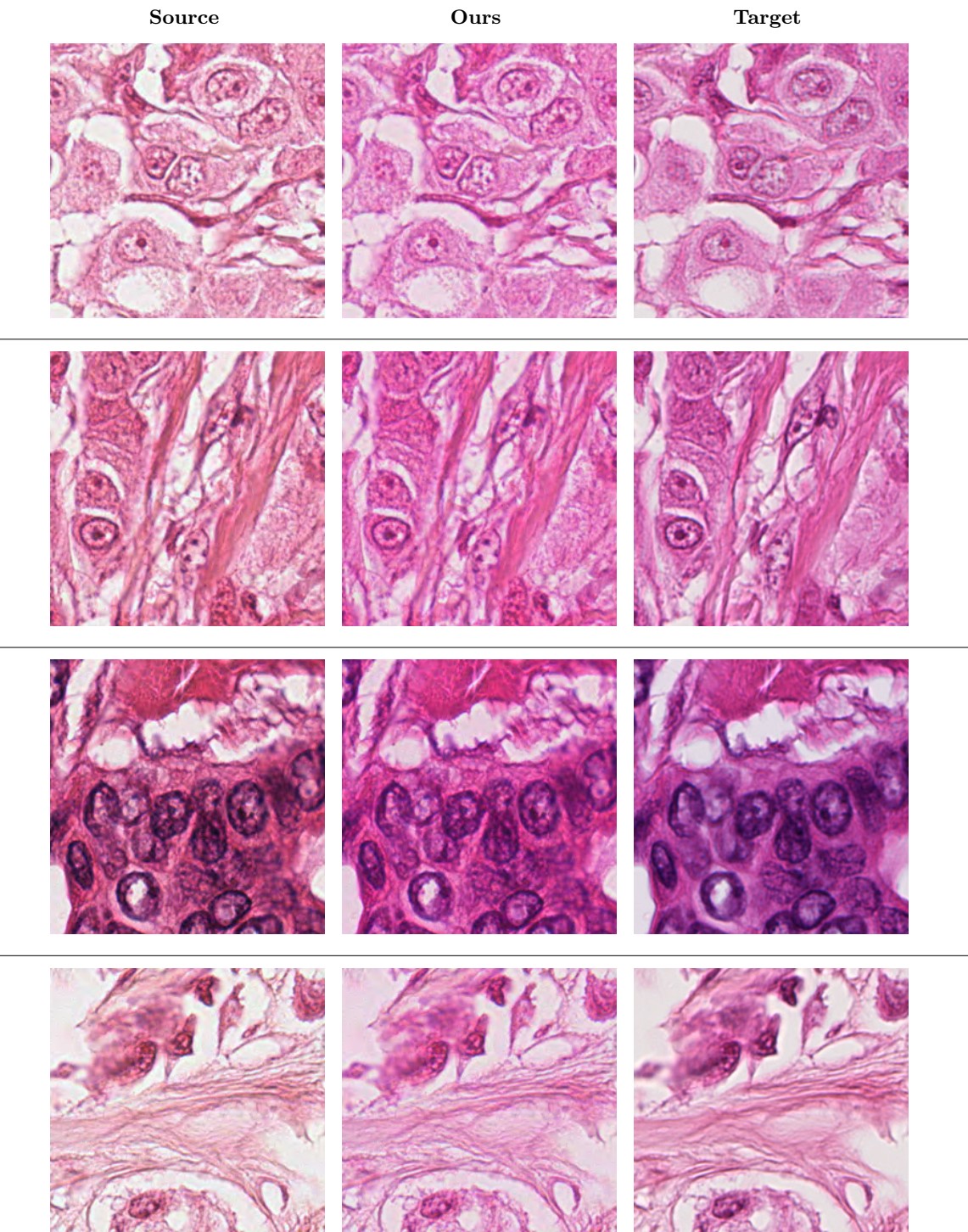

Figure 15: Evaluation on Mitos & Atypia 14 after training with heterogeneous tissue data. See section 6.3 for details.

