# OpenReview forum: "One-by-One Stainer: A Fast and Hallucination Resilient Domain Adaptation Method for Histopathology"
_MIDL.io/2026/Conference — MIDL 2026 Poster_

### Official Review · Reviewer_GxQe · 2026-01-07

**Confidence:** 4
**Preliminary Rating:** 2
**Final Rating:** 2

**Summary:**

The manuscript describes an unsupervised stain normalization method that reduces hallucinations. The authors have used the target color distribution in a similarity loss term to inform the model of the distribution to match. The network architecture is designed to keep transformations minimally non linear to prevent hallucinations.

**Strengths:**

- The 1x1 convolutions, inclusion of skip connections and weight regularization are simple but effective methods to enable near-identity behavior to ensure no artifacts/hallucinations are created as a result of color normalization.
- The small network size also helps with computational efficiency.
- Including a similarity loss to match distributions is a novel idea.

**Weaknesses:**

- The method takes the global target distribution without considering local morphological differences. Additional comments on how results would look on images with differing local distribution but similar global distribution would be interesting.
- The authors mention they have injected “just enough” non linearity in the network. Explaining how they arrived at what is “enough” or explaining failure modes would be useful.
- The results section does not offer comparison with the same set of baselines. 1x1 Stainer is compared against different sets of baselines for similarity metrics, downstream performance and hallucination quantification. Comparing against a consistent set of baselines or explaining why only a subset of baseline methods are used for each of the results would offer further clarification.
- Given comparable performance to baselines, the authors should more clearly state the advantages of the proposed work.
- The manuscript can be organized better. Details on “properties”, “ablation studies” should be separate sections.

**Detailed Comments:**

The authors may have suggested that lower similarity metrics might be because artifacts like blur are not being reproduced in their method (Sec 4.1 “It may not be desirable to perfectly reproduce these variations.”). This claim would be more convincing with some images.

**Justification Of Final Rating:**

I thank the authors for their time in responding to my comments. I am unfortunately unable to revise my original rating, as some of my original comments have not been fully addressed. While the authors provide thoughtful discussion of their design choices and clarify certain architectural decisions, several methodological concerns remain unresolved. In particular, questions regarding local versus global distribution mismatches and the use of inconsistent baselines across experiments persist, which limits the clarity and overall strength of the evaluation.

**Justification Of The Preliminary Rating:**

The authors have worked diligently on an important and relevant topic. An effective domain adaptation method can greatly improve workflow efficiency. There is novelty in using a simple architecture with a loss term that directly ties with the target distribution. Clearly definitely what the gap areas in performance are and more/better comparisons with baseline would improve the overall quality of their work. Given that the results are, at best, comparable to existing methods, the authors could explain the benefits of using the proposed work with better clarity.

**Questions To Address In The Rebuttal:**

Already mentioned above.

---

> ### Author Response · Authors · 2026-01-25
>
> We sincerely thank the reviewer for their careful reading and constructive feedback.
> In particular, it is encouraging to receive recognition of the importance and relevance of the stain normalization problem, as well as the potential impact of effective domain adaptation on clinical and research workflows.
> We are grateful for the positive assessment of our design choices aimed at minimizing hallucinations through near-identity transformations, as well as the acknowledgment of the novelty of incorporating a target distribution–based similarity loss.
> The reviewer’s suggestions regarding architectural non-linearity, baseline consistency, result presentation, and clearer articulation of practical advantages are highly valuable, and we will address them thoroughly in the rebuttal and revise the manuscript to improve clarity and organization.
>
> # Capacity and non-linearities
> First, we agree that the choice of words is vague and we corrected this accordingly in section 4.1.
> As a starting point, we took what others, i.e. the authors of StainNet, had found was sufficient for stain normalization.
> Your question is of course a relevant one, and we had explored it ourselves too.
> The capacity for non-linearity that the network needs is dependent on the differences between the source and target domain, and how well the train sets represent these.
> To assist the reader in building an intuition, we have added results on camelyon16 for 1-layer and 6-layer versions of our network, see ablations 6.2.
> A network with only 1 layer also performs quite well. For the sake of proposing a general architecture that also performs well on more challenging benchmarks, we opted for the prior work inspired 3-layer network.
>
> # Differing training set distributions
> This is indeed the main hurdle for applying our method. Thank you for stressing its importance.
> Our method closes a large part of the domain gap by modeling the differences in color digitization between two scanner pipelines.
> If the color distributions of the source and target train sets are very similar, the regularization will keep the model from straying far from the identity function.
> For this to be incorrect, the semantic meanings of colors would need to be flipped between two scanning pipelines.
> Furthermore, these swapped colors would need to be sufficiently distinct to impact downstream tasks.
> Regardless of how this would happen, in such a scenario, it is true that a color flow learned through distribution matching would not be an appropriate tool for domain adaptation.
>
>
> On the other hand, if the images in the source and target train sets do not even contain the same true colors, this modelling will fail.
> However, if these sets are sufficiently varied and the relative occurrences of true colors are close enough, a meaningful color flow can be learned.
> Due to the nature of neural network optimization and the use of weight decay, it is hard to quantify the required diversity for this to work.
> Instead, in section 6.3, we show that even when training on a heterogeneous set of tissue types (lung, uterus, pancreas, skin, brain, excluding breast), the resulting test breast outputs still closely resemble their target pairs.
>
> # Advantages of the proposed work.
> Communicating the importance of hallucination resilience and the advantages of our method, was one of the main challenges while writing down our findings.
> Our intention is for the table on the first page to clearly summarize the advantageous characteristics, followed by an entire section describing these characteristics.
>
> To illustrate a hurdle for surpassing related works on benchmarks, we have added Figure 12 to the appendix showing some of the images that were misclassified in camelyon16.
> When gathering images that are misclassified through our method but are correct when processed by some other methods, a noticeable trend appears.
> Images that contain acquisition artifacts seem more likely to lead to misclassifications.
> This is because of the design of our method.
> The anomalous pixels of the artifacts are retained after normalization, while other methods transform these toward what's expected.
> In our opinion, this altering of rare pixels is outside the range of what a normalization or domain adaptation model is trained to do.
> The potential marginal downstream gains from allowing such models to modify anomalous pixels and patches to what's expected does not outweigh the risk of masking out rare clinically relevant information.
>
> # The manuscript restructuring
> Thank you for your feedback.
> We have changed the structure of the text to reflect your remarks and improve the reader experience.

---

> > ### Author Response · Authors · 2026-01-30
> >
> > Dear Reviewer GxQe,
> >
> > As Area Chair Ye1Z kindly reminded us, only a few days remain in the discussion period. We would like to respectfully ask you to review our response for any remarks we may have overlooked. It is our full intention to address all your concerns to further improve and clarify our submission.
> >
> > As highlighted in our Rebuttal Supporting Material, we have added, among other improvements, additional results on more diverse datasets and included performance comparisons for architectural variants.
> >
> > If our response and revisions adequately address the identified weaknesses, we would be sincerely grateful if you could consider updating your score.
> >
> > Thank you once again for your time, effort, and valuable feedback.

---

### Official Review · Reviewer_99jp · 2026-01-08

**Confidence:** 2
**Preliminary Rating:** 3
**Final Rating:** 4

**Summary:**

This paper proposes 1*1 stainer, which is a lightweight, unsupervised stain normalization for histopathology images. Key idea is to strictly constrain the model architecture to pixel-wise colar mappings with residual connections, thereby preventing spatial hallucinations while still allowing limited non-linear color adaptation. The model is trained without paired data, adversarial objectives, or cycle consistency, using a novel color distribution dissimilarity loss based on projected optimal transport.

**Strengths:**

- The proposed method is simple and lightweight, without the need of paired training data.
- The paper is generally written well and the visual results look promising.
- No paired data, no GAN pretraining, no cycle consistency, and no diffusion models. This makes the method significantly easier to train and deploy than many baselines.

**Weaknesses:**

- Because the method operates purely at the pixel level without spatial context, it cannot correct domain shifts arising from texture or scanner-induced structural artifacts. While this is an intentional design choice to avoid hallucinations, it restricts applicability to domains where color differences dominate.
- The authors should involve GT for visual results whenever available.

**Detailed Comments:**

- I think the motivations behind the choice of deal with pixels independently need to be further demonstrated.
- I am not an expert in the histology images, and the authors used  a dataset contains coregistered images from two different domains, but I did not find visual results of the proposed method and ground-truth target for this dataset.
- My major concern is, I am not sure that operating only on pixel level could cause any implications for the adapted output images. I would adjust my score based on retutals.

**Justification Of Final Rating:**

I am happy with the authors' replies, which address my concerns. I will therefore increase my rating towards acceptance. However, as I am not an expert in Histopathology, my confidence score remains the same.

**Justification Of The Preliminary Rating:**

This paper proposes a very simple method to achieve better domain adaption performance over GANs, diffusion models. My major concern is, I am not sure that operating only on pixel level could cause any implications for the adapted output images. I will adjust my score based on rebuttals.

**Questions To Address In The Rebuttal:**

- The proposed method is pretty easy and lightweight. However, it deals with images pixel by pixel. In this case, it will not look at nearby pixels for a specific pixel. Will this affect the resulting output images? If not, what is the clinical explanations behind it?
- Does number of layers of the small neural networks matter? How did you decide it is 3 layer?
- I believe involving GT target images whenever available would be helpful.

---

> ### Author Response · Authors · 2026-01-25
>
> Thank you for your thoughtful and constructive feedback.
> We particularly appreciate the recognition of the practical advantages of our method, including its simplicity, lightweight design, and lack of reliance on paired data, or cycle consistency, which indeed make it easier to train and deploy in real-world settings.
> We would also like to thank you for engaging deeply with the design choice of pixel-wise transformations and for highlighting both its strengths and limitations.
> The questions and suggestions regarding the implications of pixel-level operation, architectural depth, and the inclusion of available ground truth visual comparisons are very helpful, and we will address them carefully in the rebuttal and revise the manuscript accordingly.
>
> # Pixel by pixel approach
> Yes, in a way it will affect the output images, in the sense that the texture of the output images will still be the same as those of the input images.
> The authors of StainNet before us have already shown that a pixel by pixel approach can be effective for stain normalization.
> The only potential issue with this approach is that of color collapse, where multiple colors are mapped to the same output color, without any consideration of the surrounding pixels.
> In Figures 4b and 10c we show that this indeed occurs in StainNet.
> Through our contributions to the architecture, by making it residual, and adding weight regularization during training, we avoid this issue.
> Additionally, we propose a more straightforward training scheme than the one used to obtain StainNet.
>
>
> As an intuitive explanation, the aim of our method is to model the differences between scanning pipelines.
> In that sense, the color to color function that is learned by our network can be seen as an approximation of the composite of first mapping the input RGB to a true color class, as if you were to look at the slide physically,
> followed by the mapping from the true color to the digitized color of the target scanner.
> As the colors detected by scanners are also barely influenced by surroundings, a pixel by pixel approach can lead to a good approximation.
>
> # Network depth
> In theory, fully convolutional neural networks are universal approximators.
> However, in practice, the capacity of the network matters, as you rightfully point out.
> Although we knew it would be insufficient, we actually started from invertible linear models and gradually increased the capacity.
> The 3 layer approach gave good results in the varied settings of camelyon16, mitos \& atypia 14, midog 21, and with our proprietary data.
> Naturally, the ideal setting of hyperparameters is also dependent on the architecture.
> One could further increase the network depth by including more residual connections, similar to the original ResNets.
> Although nothing major, we seemed to observe a slight trade-off wherein convergence becomes slower with more capacity.
> We have added a subsection in the ablation and table rows to aid the readers in building an intuition for finding the right capacity.
>
> # Comparisons to GT images
> You are absolutely right that qualitative comparisons to target images make the paper more accessible to readers.
> In the revised version, we make better use of the paired data in Mitos & Atypia 14 and added Figures 12, 13 and 15 to the appendix.

---

> > ### Author Response · Authors · 2026-01-30
> >
> > Dear Reviewer 99jp,
> >
> > As Area Chair Ye1Z noted, the discussion period is nearing its end. We would appreciate it if you could take a moment to review our updated response.
> > Should you wish to discuss the pixel‑by‑pixel approach in more depth, it would be our pleasure to elaborate further.
> >
> > If our clarifications resolve your concerns, we would be grateful if you might consider updating your score.
> > Thank you again for your thoughtful feedback.

---

> > > ### Comment · Reviewer_99jp · 2026-01-31
> > >
> > > I am happy with the authors' replies, which address my concerns. I will therefore increase my rating towards acceptance. However, as I am not an expert in Histopathology, my confidence score remains the same.

---

### Official Review · Reviewer_mu6W · 2026-01-12

**Confidence:** 3
**Preliminary Rating:** 3

**Summary:**

This paper addresses the problem of domain adaptation in histopathology images. Authors observe that the problem here is mainly about color/stain normalization & this propose a 1x1 convolutional network to perform pixel-wise color mapping without any spatial context. This prevents hallucinations of structures which is a limitation of prior approaches.

Moreover, the method avoids adversarial training and instead uses an approximation of optimal transport to match color distributions. The paper demonstrates that this approach achieves competitive performance while maintaining structural integrity and being much faster than GAN-based methods.

**Strengths:**

- The authors provide a compelling comparison using a proprietary dataset specifically annotated for hallucinations (Figure 1 and Figure 3), effectively demonstrating the failure modes of competitor models.


This works makes several important architectural contributions
  1. The use of 1x1 convolutions to ensure pixel-wise transformations is a clever architectural choice that directly addresses the hallucination problem. Avoiding hallucinations by design is the strongest aspect of this work.
  2. The second interesting aspect of this work is avoiding adversarial training & not needing paired datasets.


The paper is well-written and presents a very practical solution to a specific problem. The shift away from over-parameterized GANs back to constrained, interpretable models is interesting.

**Weaknesses:**

- While authors leverage 1x1 convolutions to prevent hallucinations, the architecture is limited in expressiveness. The work relies on the assumption that there are no spatial staining artifacts, which may not always hold true in practice. This limits the method's ability to correct for spatially dependent staining issues.
- The distribution matching loss assumes that the target domains have similar image content (e.g., similar tissue types). If the target batch is not representative of the overall distribution (e.g., a batch of mostly background slide vs. a source batch of dense tumor), the mapping could be skewed. This should be discussed in more detail. The sensitivity to batch composition during training should be discussed as well.

**Detailed Comments:**

I appreciate the clarity of the writing and the practical focus of the paper. It is interesting to see a simple color matching approach that is competitive with more complex GAN-based methods. The paper is well-structured and the figures are informative.

Specific Technical Questions:

- $SSIM_{src}$ -- Is this measure the same structure discrepancy metrics mentioned in sec 4.3? If so, please clarify this in the text. If not, please clarify what exactly $SSIM_{src}$ is measuring, how it is measured and how it differs from the structure discrepancy metric.
- $N=8$ in Equation 1: Did you ablate this hyperparameter? Does increasing $N$ improve color convergence significantly, or is 8 sufficient?
- Why does StainGAN & StainNet have better AUC scores despite hallucination?
- What would happen if StainGAN or stainNet are trained with 1x1 convolutions?

**Justification Of The Preliminary Rating:**

The paper shows a simple 1x1 conv architecture with color matching loss can align histopathology images. Training is non-adversarial & doesn't need paired images. Author demonstrate that other GAN based method with more expressive architectures can lead to hallucinations.

The proposed approach doesn't outperform prior approaches. Downstream task accuracy is lower despite better visual results.  This could mean that this approach have lost some crucial information from images during adaptions. I would like to hear author's clarifaction about this.

Overall, I feel paper show that a simple color matching approach can come close to state of that art which could be an intersting result for the community, so I lean towards a weak accept.

**Questions To Address In The Rebuttal:**

- Why does StainGAN & StainNet have better AUC scores despite hallucination?
- How would the approach work when the target & source distribution have different image content? Can it be tested when this approch is applicable & when not?

---

> ### Author Response · Authors · 2026-01-25
>
> Thank you for your review. It was a pleasure to read your concise summary of the main message of our work. The weaknesses you pointed out are indeed valid. We acknowledge that spatial staining artifacts can be part of the domain gap. However, our assumption is that a color function will already address the bulk of the gap. The improvements that we achieve with our normalization support this assumption. In future work, the focus can shift to hallucination-resilient spatial adaptation, perhaps as a step after color mapping.
>
> The dependence on the representativeness of the train sets is indeed one of the main weaknesses of our method.
> As a further exploration of this limitation, we added an experiment in section 6.3 where we train on a heterogeneous set of tissue types (lung, uterus, pancreas, skin, brain) and test on the unseen breast tissue type.
>
> # Technical clarifications
> - SSIM_{src} is a different measure from Structural Discrepancy.
>     It measures images quality whereas SD is meant for artifact or hallucination detection.
> - The number of random orthonormal bases N is a hyperparameter similar to the mini-batch size in stochastic gradient descent.
>     The available VRAM was the main consideration for choosing its value.
> - StainNet is in fact trained with only 1x1 convolutions. It does not have the residual connection or the weight regularization.
>     Ablations on those improvements were included in the paper.
>
> # StainGAN & StainNet difference in AUC
> We understand your concerns regarding the performance on downstream tasks.
> In the end, that is an important measure of the usefulness of a method, and information loss is indeed a crucial argument in this discussion.
> There are a few points that we would like to make.
> One, downstream performance is not the only measure by which we should compare normalization approaches.
> In our opinion, the risk of hallucinations subtracts from the applicability of other methods.
> Two, the mean value of AUC is lower but on closer inspection, one notices that the error margins overlap, making these results statistically similar.
> Finally, following your valuable suggestion to investigate the apparent gap,
> we added a figure to the appendix showing some samples on which classifiers fail after normalization with our method but succeed with other methods.
> A noticeable trend is that these samples often contain acquisition artifacts.
> Due to the design of our method, these anomalous pixels are retained after normalization.
> The examples show that StainGAN and StainNet tend to modify these pixels toward what is expected.
> In these cases, these remaining artifacts could be the cause for distraction of the classifier and misclassifications.
> However, we argue that normalization models are not trained to distinguish between acquisition artifacts and rare clinically relevant pixels.
> Models that exhibit this type of color collapse effectively suffer from information loss.
> The marginal downstream gains from smoothing out the inputs do not outweigh the risk of hallucinations.
>
>
> # Target & source distribution different image content
> That is a very relevant question.
> We are happy to report that it can work and we added section 6.3 and Figure 15 in the appendix to illustrate this.
> For that experiment, we gathered a training set of heterogeneous tissue types (lung, uterus, pancreas, skin, brain) that importantly did not include breast tissue.
> We then trained a model to normalize to the Mitos & Atypia 14 target domain that consists of images of breast tissue, and show qualitative comparisons with test set data.
> Of course, there are limits to this. Because of the 1x1 field of view, it is not so much the image content but rather the relative occurrences of colors that determine the success.
> As long as the relative occurrences of true colors, before digitization, in the source and target sets are close enough, a meaningful color flow can be learned.
> A benefit of the weight regularization is that these occurrences don't have to match exactly.
> The loss function with this regularization term, encourages a smooth color mapping.
> On the other hand, because of this regularization and due to the nature of neural network optimization, it is hard to give a quantitative guide for applicability.

---

> > ### Author Response · Authors · 2026-01-30
> >
> > Dear Reviewer mu6W,
> >
> > Following Area Chair Ye1Z’s reminder about the remaining discussion time, we kindly ask you to take a moment to review our updated response.
> >
> > As described in the Rebuttal Supporting Material, one of the improvements is an additional investigation comparing downstream classification performance between our method and StainGAN. We also added results after training with different tissue types in the source and target sets.
> >
> > If these additions resolve your earlier concerns, we would sincerely appreciate a reconsideration of your score.
> > Thank you for your time and valuable comments.

---

### Author Rebuttal · Authors · 2026-01-25

**Rebuttal:**

Incorporated remarks from reviewers. These changes are highlighted in green in the text.

**Supporting Material:**

/attachment/10d907701e076b13b024a6b7c54981a0835f49c4.pdf

---

### Comment · Area_Chair_Ye1Z · 2026-01-28

Dear reviewers,

The authors have now responded to your reviews. At this time please participate in discussions with the authors.

IMPORTANT: You must enter your final rating by clicking “Edit” → “Official Review” and providing the Final Rating by February 1st 2026 (23:59 AoE).

Thank you again for your service to MIDL 2026 and making it a success.

---

### Comment · Area_Chair_Ye1Z · 2026-02-01
**final ratings**

Dear reviewers, if you have not done so already please provide your final ratings for the paper today. If you already did, please disregard this message.
Thanks!

---

### Meta-Review · Area_Chair_Ye1Z · 2026-02-09

**Recommendation:** Accept (Poster)
**Confidence:** 4

**Metareview:**

The reviewers found the 1x1 convolution approach to stain normalization for avoiding hallucinations and the data distribution matching to be novel and practically useful. One reviewer officially changed their score from borderline to weak accept after the rebuttal. Another one mentioned also changing from borderline to weak accept after the rebuttal in their justification but did not officially change their score. I am considering their score as weak accept since they explicitly state this in their text. The third reviewer found several issues including the limited evaluations and inconsistency in the use of baseline methods/datasets. Their rating is weak reject. I agree with this concern which does not appear to be addressed in the rebuttal. It is not clear why one dataset if used for downstream evaluation and another is used for stain transfer evaluation. I find this to be an important weakness of the paper, and my overall impression is of a borderline paper. However, in light of the novelty and potential practical usefulness of the method, I would still recommend the paper to be accepted only if there is room in the program.

---

### Decision · Program_Chairs · 2026-02-13

Accept (Poster)